# Consumer Preferences and Socioeconomic Factors Decided on Plantain and Plantain-Based Products in the Central Region of Cameroon and Oyo State, Nigeria

**DOI:** 10.3390/foods10081955

**Published:** 2021-08-22

**Authors:** Patchimaporn Udomkun, Cargele Masso, Rony Swennen, Bhundit Innawong, Apollin Fotso Kuate, Amos Alakonya, Jules Lienou, Dorcas Olubunmi Ibitoye, Bernard Vanlauwe

**Affiliations:** 1International Institute of Tropical Agriculture (IITA), Bujumbura 1893, Burundi; 2IITA, Yaoundé BP 2008, Cameroon; C.Masso@cgiar.org (C.M.); A.Fotso@cgiar.org (A.F.K.); J.Lienou@cgiar.org (J.L.); 3IITA, Kampala 7878, Uganda; R.Swennen@cgiar.org or; 4Department of Biosystems, KU Leuven, 3001 Heverlee, Belgium; 5Department of Food Technology, Faculty of Engineering and Industrial Technology, Silpakorn University, Nakhon Pathom 73000, Thailand; b.innawong@gmail.com; 6International Maize and Wheat Improvement Center (CIMMYT), Texcoco 56237, Mexico; A.Alakonya@cgiar.org; 7National Horticultural Research Institute (NIHORT), Ibadan 5432, Nigeria; bunmiajisafe@yahoo.com; 8IITA, Nairobi 30772-00100, Kenya; B.Vanlauwe@cgiar.org

**Keywords:** banana and plantain, consumer behavior, consumer choice, processed products, quality attributes, willingness to purchase

## Abstract

Plantain is a key staple food in Central and West Africa, but there is limited understanding of its market in Africa. In addition, the cooking methods for enhancing the nutritional value, consumer preference, and willingness to pay for plantain and plantain-based products are not well understood. The knowledge gaps in the market and consumer dimension of the food chain need to be known to increase plantain utilization and guide breeding efforts. This research contributes by examining the cooking methods, consumer preference, and willingness to pay for plantain and plantain-based products in Cameroon and Nigeria. A household survey sample of 454 Cameroonian consumers in four divisions of Central Region and 418 Nigerian consumers in seven government areas of Oyo State in southwest Nigeria was the basis for the analysis. The results showed some levels of similarity and difference in the consumption and cooking of boiled, roasted, and fried plantain in both countries. The trend in consumption of all plantain-based products was constant in Cameroon but increased in Nigeria. The most important factor influencing Cameroonian consumers’ choice of plantain and its products was taste, while the nutrition trait influenced Nigerian consumers. Both Cameroonian and Nigerian consumers considered packaging, location of produce, and size and quantity as the least important factors. In addition, socioeconomic characteristics were significant determinants of consumers’ choices to consume plantain and its products. Gender significantly influenced (*p* < 0.05) taste, while nutrition was significantly driven (*p* < 0.05) by education and annual income. Household size played a significant role (*p* < 0.05) in consumers’ choices when the price was considered. These findings serve as a guideline to improve existing products to match the needs of consumers in each country and develop products for different consumer segments and potentially increase production.

## 1. Introduction

Plantains are considered a good source of starch and energy in sub-Saharan Africa (SSA) [1] and are nutrient-rich in dietary fiber, minerals (potassium and phosphorus), vitamins (A, B1, B2, B6, and C), and phenolic compounds [2,3,4,5,6]. Unripe plantain has been documented as a hypoglycemic plant used in the management of diabetic complications [7]. Ripe plantain contributes significantly to food security and provides a daily intake of more than 25% of carbohydrates and 10% of calories for more than 70 million people in sub-Saharan Africa [8,9]. However, the availability and concentration of those chemical constituents vary with the cultivar, postharvest maturation, growing location, climate, and agricultural practices [10,11,12]. Apart from playing an important role in addressing food security for over 500 million people in the SSA region, plantain also contributes to the subsistence economies in Africa [13] and generates considerable employment [14]. Marriott and Lancaster [15] reported that plantain cultivation is more lucrative to farmers than cassava, maize, rice, or yam due to lower labor requirements. The main plantain producing regions are Central (42%) and West (40%) Africa, accounting for approximately 60% of worldwide production [16]. During 2010–2019, plantain production increased by 36.7% in Central Africa and 20.8% in West Africa. The highest plantain-producing countries in Central Africa are the Democratic Republic of Congo and Cameroon, whereas Ghana and Nigeria are two major producers in West Africa.

As postharvest maturation affects the ripening rate, plantain is mostly harvested green at different maturity stages depending on the distance to markets [17]. Depending on the consumer, plantain is eaten fresh at the ripened stage or is locally processed into different dishes, depending on the ripening stage [18] and the custom of the ethnic group [19]. In Cameroon, plantain plays an important role in family events such as weddings, funerals, and the celebration of a newborn child, particularly in the South. Sometimes, the plantain in a meal is compulsory [20,21,22]. Plantain is also a main ingredient in Nigerian cuisine; Nigerians often prefer plantain for breakfast [23,24]. In addition, commercial production of plantain-based products such as chips and flour has been developed in both countries [20]. Data from FAOSTAT [16] shows that the quantity of plantain for processing increased tremendously by 382% in Cameroon during 2003–2013. Adeoye and Oni [25] linked the increase in demand for raw plantain in Nigeria to more local processing industries.

Although the current demand for plantains and their products is high, consumers’ needs and tastes change over time [26]. Understanding consumers’ trait preferences is the first step in developing a demand-driven breeding program. In addition, knowledge of traits that consumers prefer will enable researchers and farmers to produce marketable cultivars with acceptable attributes. In the case of producers, it is critical to know what consumers like or dislike about existing products [27] to innovate a product successfully. In general, consumer preferences are conditional on both intrinsic and extrinsic factors [28,29,30]. Li et al. [31] explained that intrinsic factors are those that cannot be altered without physically changing the composition of the product, such as color, aroma, texture, and so on. Extrinsic factors can influence consumer choice and include the brand, price, packaging, label claims, perceived satiety, emotional response, decision-making process, and the emotional impact often influenced by culture. Therefore, actors in the plantain value chain who understand the factors influencing consumer preferences will be able to identify the impact on consumers’ purchasing decisions [32,33,34], particularly when food products are becoming more complex owing to consumers’ demands [35]. There is evidence that producers who engage in value addition in agricultural value chains are often faced with competition from experienced competitors who have established networks of consumers [36].

However, scientific information about consumers’ needs and preferences of plantain and their products is still limited, as few studies have been reported. A better understanding of such information is needed to enhance the adoption, production, and utilization of plantain products in Cameroon and Nigeria. In this paper, therefore, the characteristics of Cameroonian consumers in the Central Region and Nigerian consumers in Oyo State regarding their perception towards plantain and plantain-based products were examined to seek ways to improve the quality of existing products and develop new ones. In particular, the study evaluated the consumption characteristics and investigated the effect of sociodemographic attributes or factors that influence consumers’ choices for plantain and its products.

## 2. Materials and Methods

### 2.1. Sampling and Survey Design

A consumer survey was conducted between April and May 2020 in rural, peri-urban, and urban areas of Cameroon and Nigeria. In Cameroon, the Central Region was selected, while Oyo State represented the study area in Nigeria. As regions of Cameroon are in divisions and subdivided by districts, the selected divisions in Central Region were Mfoundi (districts: Yaoundé 1 and Yaoundé 6), Nyong et So’o (district: Mbalmayo), Nyong et Mfoumou (districts: Ngoumou, Akonolinga, and Ayos), and Mbam et Kim (district: Ntui) (Figure 1A). In Nigeria, each state is divided into local government areas (LGAs) and then subcommunities. The selected LGAs in Oyo State were Ona Ara (subcommunities: Olunloyo, Amuloko, Gbedun, and Akanran), Akinyele (subcommunities: Alabata, Ijaye, Ojedeji, Saanu, and Oladele), Egbede (subcommunities: Monatan, Olodo, Alakia, and Osegere), Ibadan-North East (subcommunities: Idi-Ape, Iwo, and Bashorun Oja), Ibadan-North (subcommunities: Yemetu, Oke Are, and Sango), Oluyole (subcommunities: Abanla, Idi Ayunre, and Elebu/Orita), and Ido (subcommunities: Ologuneru, Ile Tuntun, Apata/Odo Ona, and Elewe) (Figure 1B).

A stratified multistage sampling procedure was used for data collection. In the first stage, region/state areas with high plantain production and consumption were listed; subsequently, districts/local government areas were randomly selected. In the second stage, the selected districts/local government areas were divided into blocks (production and processing/marketing areas), and blocks were randomly chosen from within each selected collection district/local government area. Within each selected block, a list of households was generated in the third stage, and some of them were retained based on probable proportional size. Lastly, a total sample was picked based on systematic sampling until the actual study of total households was sampled. After selection, face-to-face interviews with respondents were conducted by trained enumerators using a semistructured questionnaire administered in English and French according to their language of choice.

Sociodemographic data was first collected from the respondents on area classification, gender, age, marital status, year of education, education level, household size, number of women and children below 5 years of age in household, annual household income, main sources of household income, household expenses, household expense for plantains, purchasing locations, and major sources of fuel and water supply for cooking.

From the survey data, the majority of Cameroonian respondents in the Central Region (74%) and Nigerian respondents in Oyo State (58%) lived in urban areas (Table 1). Overall, more females (62%) responded than males with an average age of 40 years but varying between 18 to 90 years. The average household size was 7 persons in Cameroon and 5 in Nigeria, with approximately 40% of females and 16% of children below 5 years of age in households. About 59% of Cameroonian and 73% of Nigerian respondents were married and living with their spouses. The average year of education of respondents was 11.2 in Cameroon and 13.6 in Nigeria, with more than 40% of the respondents having secondary school certificates, while less than 15% of them had a university education. The total household annual income of most Cameroonian respondents (47%) ranged between USD 1801 and 6000, while more than half of Nigerian respondents earned a household annual income that ranged from less than USD 600 to 1800. The main source of household income in Cameroon was crop sales (30%), and in Nigeria, craftmanship (32%). About half of household income was spent on food, and a third of that household food expenditure was used to purchase plantain. Gas was the main fuel for cooking in both Cameroon (45%) and Nigeria (54%). In Cameroon, 38% of the respondents received water from pipes for cooking, while 52% of Nigerian respondents used water from wells.

Subsequently, the respondents were asked about consumption frequency, consumption trends until now, cultivar commonly used, ripening stage used, cooking ingredients, common accompaniment, and cooking time of each plantain-based product (boiled, roasted, fried as chips, fried as *dodo*, flour, pounded, and porridge). Thirdly, the respondents were asked to rank the importance of certain factors (nutrition, food safety, taste, appearance, availability, price, size and quantity, packaging, advertisement, location of produce, and knowledge) in their choice to consume plantain-based products (Always = 1; Sometimes = 2; Never = 3). Specifically, “food safety” was explained to the respondents as “food (plantain-based products) that will not harm anyone when it is prepared and/or eaten in accordance with the food’s purpose”. Later, the respondents were asked, “Have you ever purchased plantain products from the market?” If the respondents said “Yes”, the enumerators continued by asking questions relating to their satisfaction with the products and main factors influencing their willingness to pay for new plantain products.

### 2.2. Data Analysis

STATA software version 12.0 (StataCorp. LP, College Station, TX, USA) was employed to analyze the data. Descriptive statistics were used to determine the means and frequencies of respondents’ replies regarding collected data. Each ranked factor mean was computed to identify the most important factor. Chi-square (*χ*^2^) and analysis of variance (ANOVA) were used to examine the differences in response from consumers. The Duncan Multiple Comparison Test (*p* < 0.05 confidence levels) was used for means comparison, and ANOVA tests were applied to relate the mean values with the consumers’ socioeconomic characteristics to understand the factors influencing consumer’s choice for plantain-based products.

## 3. Results and Discussion

### 3.1. Consumption and Cooking Methods of Plantain

In Cameroon, almost half of the respondents buy plantain and their products every month, while most Nigerian respondents buy plantain (52%) and their products (37%) every week (Figure 2a1,a2). Likewise, markets inside each community (>60%) were the main source for buying fresh plantain and their products (Figure 2b1,b2).

Plantain is usually prepared in different forms, such as boiled, roasted, fried, pounded, porridge, and flour, depending on the ripening stage [37]. The most commonly preferred dishes by the Cameroonian and Nigerian respondents were boiled and fried plantain as *dodo*, respectively (Table 2 and Table 3). More than 40% of the Cameroonian and Nigerian respondents consumed boiled and fried plantain as *dodo* weekly; however, other plantain-based products were mostly consumed once a month. For boiled plantain, unripe or ripe plantain fingers are peeled and cooked in salted boiling water for 15–35 min. Boiled plantain are consumed with various sauces or other accompanying dishes. Fried plantain as *dodo* is processed by peeling and cutting into slices before frying in vegetable oil for about 8–10 min. Honfo et al. [19] and Okolle et al. [22] explained that the frying method is rapid and less time-consuming, and *dodo* could also be accompanied by various sauces, vegetables, and other food complements. These results agreed with Ayinde et al. [9], who reported that boiled and fried plantain represented the most important consumption form preferred by Nigerian households.

Interestingly, more than 20% of the respondents ate roasted, pounded, porridge, and flour products during occasions such as weddings and funerals. Roasted plantain is cooked on charcoal and often sold in the street with roasted fish or roasted African plum (*Dacryodes edulis*). Pounded plantain, in general, is prepared by boiling plantain for 30–45 min. The pulp is then pounded in a mortar and mixed with salt and other ingredients such as palm oil (Cameroon) or yam (Nigeria) until the paste becomes homogeneous. In some households, no ingredients are added to the pounded pulp. This pounded plantain can be eaten with various sauces, depending on culture and consumer preference. Dury et al. [20] stated that pounded plantain is not a common dish in cities, but is rather a favorite for the Beti people, a Southern ethnic group in Cameroon. To prepare porridge, plantain pulp is cut and boiled with water. Subsequently, salt, soy sauce (magi), pepper, palm oil, dried fish/meat, onion/tomato, etc., are added. The entire mixture is allowed to boil for about 30–70 min, depending on the cultivar and ripening stage of the plantain. In Cameroon, groundnut and meat might be added during boiling; therefore, no food accompaniments are required, while plantain porridge in Nigeria is frequently eaten with beans, vegetables, meat, and stews. However, Amah et al. [24] reported that porridge is more important in the South-South than in Nigeria’s South-West.

In Nigeria, plantain chips are consumed as a snack, while some Cameroonians consume chips with other accompaniments such as beans, egg, fish, and meat. Plantain chips are prepared by frying slices of unripe or slightly ripe plantain pulp in vegetable oil for 10 min at a temperature of 160–180 °C. The production of plantain chips in Cameroon, Nigeria, and other African countries is principally a female activity. The consumption of plantain flour was only found in Nigeria. Honfo et al. [19] indicated that plantain flour is not incorporated in the food habit of Cameroonian people, except some from the North-West of the country. The same consumption trend of plantain flour was reported by Newilah Ngoh et al. [21]. To make plantain flour, unripe plantain is peeled and cut into small pieces before sun drying for 2–3 days. Subsequently, the dried pulp is ground in a wooden mortar or a maize grinder. A dish from plantain flour in Nigeria is prepared by mixing plantain flour with cassava or yam flour. Then water and ground ingredients of garlic, onion, groundnut oil, fish, and meat are added to the mixed flour; all ingredients are cooked for about 20 min. Boiled plantain, yam, or sweet potato together with various soups appear to be the most common food complements for this dish.

The consumption of all plantain-based products over the last 5–10 years was constant in Cameroon (~40%), while the consumption increased in Nigeria (>50%), particularly for *dodo* (74%) (Table 2 and Table 3). Both small and large plantain fingers were commonly used to prepare all these dishes; however, large fingers were preferred by Nigerian respondents for boiling, *dodo*, and pounded plantain. A finding of Ayinde et al. [38] supported that Nigerian consumers preferred plantain with medium or large fingers and with a shelf-life of 7–9 days under natural conditions. In contrast, Dury et al. [20] reported that Cameroonian women do not consider plantain a homogenous product, and usually, the preferred cultivar and size are different from one preparation to another. However, it should be noted that consumers and traders have their preferred trait preferences. For cooking bananas and plantains, consumer trait preferences are determined by the product type and processing method [20,39,40]. At the time of purchase, most consumers prefer fresh plantains with large bunches and large fruits. Kouamé et al. [37] found that for urban consumers in Côte d’Ivoire, plantain ripening/maturity stage used to prepare different foods was more important than other physical attributes. In contrast, traders only preferred plantains with large bunch sizes [41].

In Cameroon, unripe plantain was mostly used to make porridge (93%), chips (56%), and pounded plantain (43%), while boiled (47%) and roasted (41%) plantain were often prepared by using ripe plantain with a peel more yellow than green (Table 2 and Table 3). Ripe plantain with all-yellow peel (55%) and overripe plantain (34%) were used for frying as *dodo*. In contrast to Cameroonian dishes, unripe plantain (64%) was used for boiling and overripe plantain (56%) was used for making porridge in Nigeria. In addition, it was observed that Cameroonians and Nigerians use almost similar ingredients for cooking some common dishes such as boiled, roasted, and fried plantain. This assumes that people from the same culture or region have been affected in the same way culturally and therefore have similar food preferences, though it does not necessarily mean that foods need to be prepared in the same way or that they taste the same across geographical regions [42]. Another explanation of this behavior might be associated with the migration of plantain from Asia to Eastern Africa and then from Eastern Africa to Western Africa [43].

### 3.2. Factors Influencing Consumer’s Choice and Their Willingness to Purchase New Plantain-Based Products

Taste was the most important factor affecting the choice in consuming fresh plantain (66%) and their products (81%) in Cameroon (Figure 3a1,b1), followed by availability (38% for fresh plantain and 34% for plantain-based products). On the other hand, nutrition (the biochemical and physiological process by which an organism uses food to support its life) was the most positively important factor influencing Nigerian consumer’s choice in consuming fresh plantain (69%) and their products (68%), followed by taste (59% for fresh plantain and 57% for plantain-based products (Figure 3a2,b2). Many studies agreed that taste and price have more of an influence on the food choice than a product’s health benefits [44,45]. This result agreed with Kikulwe et al. [46], who reported that taste and price are the most important factors for purchasing cooking banana in Uganda. Sanya et al. [47] also documented that better consumption attributes such as taste, texture, and color could increase the likelihood of adopting improved banana hybrids by farmers in Uganda. Apart from taste, Ayinde et al. [38] observed that size and number of fingers were considered the most important factors when purchasing plantain, while appearance, color, and shelf-life were the least. Concerning taste, this finding was also in line with the choice of experiments of Hoek et al. [48], who found that when consumers find taste very important, they will not compromise and respond to price or other measures. A study of Amah et al. [24] found that most Nigerian respondents identified food quality characteristics related to color, texture, taste, and odor as the most important quality characteristics of plantain products.

In general, the presence of nutrition and health claims could negatively affect the expected tastiness of food products [49]. Verbeke [50] claimed that acceptance of foods with nutrition and health benefits has become more conditional, particularly for taste. Consumers have become more convinced that good taste and healthiness do not necessarily have to be traded off against each other. Hence, belief in the health benefits has become the strongest positive determinant of consuming foods with nutrition and health claims to such an extent that consumers are ready to compromise on taste. This reason may explain why nutrition was the most important factor for the Nigerian respondents in consuming plantains and their products. Many of them believe that the consumption of unripe plantain is good for health, especially in treating diarrhea [9]. Furthermore, Ayinde et al. [9] also revealed that the location of the respondents was an important factor in plantain consumption, as a higher percentage of urban Nigerian respondents was aware of the nutritional value of plantain, while most rural respondents were not aware. When the least important factors were considered, both Cameroonian and Nigerian respondents referred to packaging, location of produce, and size and quantity. Availability was not the most important factor, especially for the Nigerian respondents, La Trobe [51] indicated that consumers identified a lack of availability as their actual purchase barrier in purchasing locally produced goods.

In Cameroon and Nigeria, gender significantly influences (*p* < 0.05) taste, while education and annual income are associated with nutrition. In addition, household size is responsible for consumers’ choices regarding the price of plantain products (Table 4 and Table 5). Regarding the effect of gender, Amah et al. [24] reported that men primarily focus on the yield of plantain production. At the same time, women might have spread their preferences more over the other characteristics that are related to home consumption. This could be used to explain women’s dominance in plantain food product processing, involving more hands-on interaction with the quality-related aspects of the plantain. This observation also depicted that education is an important factor in increasing awareness and obtaining more information and knowledge about high-quality products. Dhamotharan and Selvaraj [52] found that the better product knowledge of highly educated consumers helps them to seek more information about labels and brands to buy quality geographical indication (GI) registered bananas. They suggested that emphasizing the importance of promoting these specific products through print and mass media by providing the information on health benefits in terms of medicinal and nutritive value could increase willingness to pay in literate consumers. Moreover, Ayinde et al. [9] displayed that consumption of plantain is determined by occupation. Their results showed that teachers with higher education consumed more plantain because of their awareness of its nutritional value. However, Garnett et al. [53] highlighted that education on healthy and sustainable food products does not necessarily shift consumer responsiveness to healthier and more sustainable foods. Hence, it is important to consider the interaction between consumers and products at the research level, and then different points of purchase should be discussed at the food policy and marketing levels.

A higher income allows better access and higher-quality food. Dury et al. [20] showed that although plantain is the preferred staple for most households, family income may constrain its consumption. A study by Kikulwe et al. [46] also reported that the Ugandan respondents with higher incomes were more likely to purchase GM bananas with improved nutrition and taste. Likewise, Jolly et al. [54] and Sabran et al. [55] stated that the wealthier are more likely to take precautions about food and are more willing to pay for high-quality products than those with lower incomes. Additionally, Silva [56] found that food-secure households with higher incomes purchased a wider variety of high-quality food items compared to food-insecure households with lower incomes. Smaller households were more likely to pay than those with a larger household size. Again, Kikulwe et al. [46] showed that larger household sizes were associated with fewer purchases of GM bananas. This also implies that larger households, especially in lower-income communities, can choose to consume only low-priced goods of lower quality. Basan [57] proposed that consumers in different income classes in the society could have their preferences when making purchase decisions concerning the embodied attributes of bananas. Therefore, a well-informed clientele can be selected, and a proper market segmentation and marketing plan can be prepared to increase the profit of farmers and traders.

Moreover, about 78% of Cameroonian and 95% of Nigerian respondents purchased plantain-based products from the market (Table 6). Although more than 70% of them were very satisfied with those plantain products, almost 100% were willing to pay slightly more for new plantain products if their quality can significantly improve their health. This result is consistent with Kim and Chung [58], who showed that health has a significant effect on consumers’ attitudes. However, Lange et al. [59] reported that consumers might indicate a strong preference or purchase intent for a product perceived as high quality without buying it when in a purchasing situation. Therefore, a more realistic assessment of price relative to other product attributes apart from health should also be considered through a conjoint study or experimental auction [30].

When the respondents were asked to indicate their attitude toward buying new plantain-based products if they were: (1) more delicious; (2) more nutritious; (3) greater quantity; (4) lower price; and (5) more available in the markets, Cameroonian respondents were willing to pay more if the new plantain products were tastier than the existing products, while new products with improved nutrition are highly targeted for Nigerian respondents. Quantity of products was considered the least when compared to other product factors. Bruschi et al. [60] stated that the willingness to pay for intrinsic attributes such as taste and nutrition also depends on other factors such as cultural and sociodemographic/economic characteristics and type of product. This agrees with Wang et al. [61], who confirmed that health concerns regarding the consumption of nutritious foods in China and Europe might be different due to dietary patterns, culture, and customs. While Köster [62] explained sensory preferences, particularly taste, as unconscious and unintentional learning processes, those established in childhood are important in predicting preferences later in life. Although the Nigerian respondents considered nutrition the most important factor influencing their choice of consumption or willingness to pay for plantain products, this can be both true or false, because most of them had little understanding of the actual nutritional value of plantain. When asked to name the important nutrients in plantain, “protein” was their first answer, or to list what kind of new plantain products they want in the markets, about 15% of the respondents listed plantain oil, products rich in protein, etc. This finding underlined that increasing consumers’ knowledge of the actual health benefits of plantain through nutrition education and training and/or spoken media is very crucial, as it might result in creating greater awareness and increasing more consumption and utilization of plantain. Zepada and Dela [63] illustrated that increasing and deepening knowledge can reinforce existing values of products, which then influence beliefs and norms, leading to sustainable food purchase behavior. In addition, Klopčič et al. [45] suggested that the functions/benefits of the nutrients need to be expressed in clear, direct, short, and simple language without using scientific terms. A symbol on a package can affect consumers’ product evaluations and stimulate purchasing [64], especially by those such as the Cameroonian respondents, who are less motivated by health.

## 4. Conclusions

This study examined consumption characteristics, cooking methods, and preferences of plantain and its products among consumers in the Central Region of Cameroon and Oyo State of Nigeria. Taste was the most critical factor for consuming plantains and their products for Cameroonian respondents, while Nigerian respondents valued their nutrition. In addition, this study clearly indicated the similarities and differences in consumption characteristics and cooking methods of plantain food products in those study areas, which can principally be explained by ethnic traditions and plantain migrations. The results also suggested that gender played a prominent role in consumers’ choice in consuming plantain-based products, while education and annual income had a significant influence on the selection of products with nutritional traits. Household size tended to be taken into consideration when the price is associated. Although a larger proportion of respondents probably buy or do not buy a certain product, it does not mean they would not buy it, especially if the new product appears on the market with distinguishing characteristics. These points are very important even if consumers are not demanding improved varieties of plantains. Breeders still need to be aware of critical end-user traits, as they provide very useful information to help plantain breeding teams to adapt to and revisit the product profiles and breeding priorities. For the food-processing sector, the physiological stages of harvesting and postharvest treatment processes should be focused on a better understanding of the dynamics of useful compounds. Investigating the various factors of plantain composition, and the interactions between the various constituents, will be a source of knowledge for the choice and control of processing operations. Moreover, innovative approaches in plantain processing; for example, using hybrid solar dryers to make high-quality plantain flour, can be applied to meet consumer demand. Finally, there is a need to create awareness of the importance of plantain with regard to its nutritional significance.

## Figures and Tables

**Figure 1 foods-10-01955-f001:**
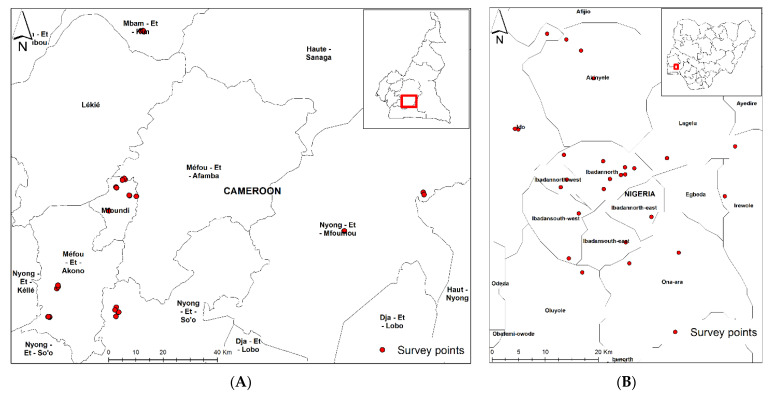
Sampling areas in Cameroon (**A**) and Nigeria (**B**).

**Figure 2 foods-10-01955-f002:**
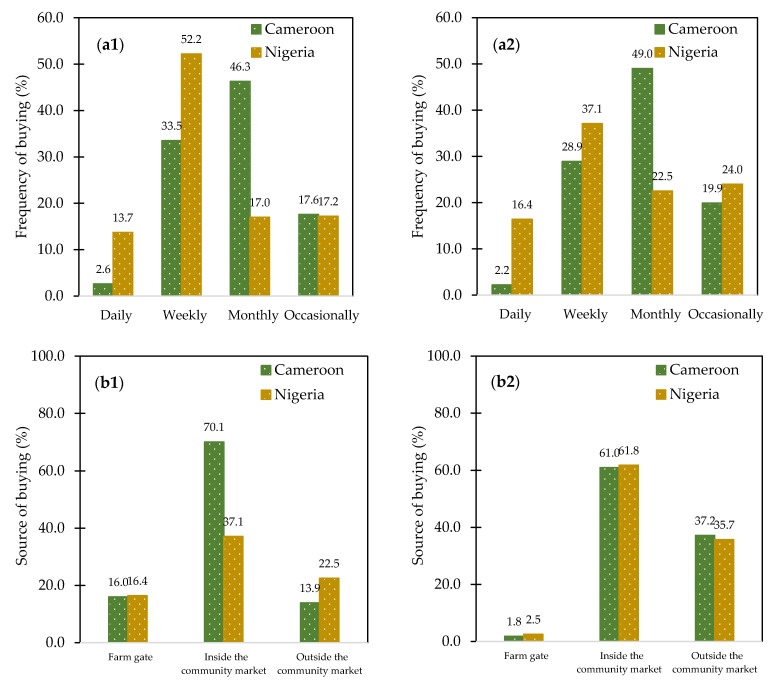
Frequency (**a**) and source (**b**) of buying fresh plantain (**1**) and plantain-based products (**2**) in Cameroon (*n* = 454) and Nigeria (*n* = 418).

**Figure 3 foods-10-01955-f003:**
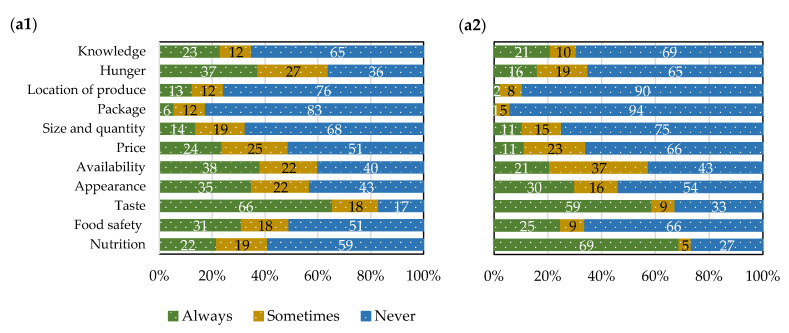
Factors influencing consumer choice for consuming fresh plantain (**a**) and their products ^1^ (**b**) in Cameroon (**1**) and Nigeria (**2**). ^1^ An average value of all plantain-based products including boiled, roast, chips, dodo, flour, pounded, and porridge; however, flour was excluded from plantain-based products in Cameroon due to unavailability.

**Table 1 foods-10-01955-t001:** Socioeconomic characteristics of the respondents.

Characteristics	Country	
	Cameroon (*n* = 454)	Nigeria (*n* = 418)
Location of the household (%)
Rural	8.8	25.1
Urban	74.4	57.7
Peri-urban	16.8	17.2
Female respondents (%)	61.8	62.1
Average age (years) ^1^	38.6 (12.7)	40.0 (14.1)
Household size (number) ^1^	6.9 (3.0)	5.4 (3.3)
Women in household (number) ^1^	3.0 (1.7)	2.1 (1.8)
Children below 5 years of age in household (number) ^1^	1.2 (1.2)	0.8 (1.0)
Marital status (%)
Never married	26.5	13.6
Married, living with spouse	59.1	72.5
Married, spouse away	2.4	4.5
Widow/Widower	6.3	8.1
Other (Engaged, Separated, Divorced)	5.7	1.3
Average education (years) ^1^	11.2 (4.5)	13.6 (5.2)
Education level (%)	
None	3.7	9.3
Primary	23.8	17.9
Secondary	41.0	47.1
Bachelor	10.1	12.2
Master/Doctorate	3.2	0.5
Other (Graduate, Religious, Adult education)	0.4	13.0
Household annual income (%)
<USD 600	5.9	13.9
USD 600–1800	41.0	51.2
USD 1801–6000	46.5	28.4
>USD 6000	6.6	6.5
Main source of household income (%)	
Crop sales	29.6	14.5
Craftmanship	27.3	32.2
Part-time labor	1.9	10.0
Permanent employment	12.8	2.5
Pension	14.7	16.1
Others (Sales of livestock, Food processing, Remittance, Others)	6.7	7.7
Household expense for foods (within 10 parts) ^1^	5.3 (1.6)	4.5 (1.9)
Household expense for plantains (within 10 parts) ^1^	2.1 (1.0)	1.6 (1.2)
Main source of fuel for cooking (%)
Charcoal	13.6	7.2
Firewood	38.6	17.6
Gas	44.5	53.8
Others (Agricultural byproducts, Electricity)	3.3	21.4
Major source of household water supply for cooking (%)
Wells	29.5	51.9
Borehole	1.3	39.2
Water conveyance	25.8	0.4
Pipe	38.4	6.0
Others (River/Pond, Others)	5.0	2.5

^1^ Value is the mean (standard deviation).

**Table 2 foods-10-01955-t002:** Consumption characteristics and cooking methods of plantain-based products in Cameroon.

Products	Boiled Plantain (*n* = 420)	Roasted Plantain (*n* = 162)	Fried as Chips ^2^ (*n* = 327)	Fried as *dodo* (*n* = 350)	Pounded (*n* = 230)	Porridge (*n* = 269)
Consumption frequency (%)						
1 time/day	0.7	1.2	1.2	0.6	0.4	0.0
>1 time/day	3.8	5.5	3.4	5.7	2.2	1.5
Weekly	50.7	29.4	39.4	43.7	28.3	27.9
Monthly	40.5	38.0	43.7	41.4	44.3	49.4
Occasionally	4.3	25.8	12.2	8.6	24.8	21.2
Consumption trend till now (%)						
Increasing	28.1	25.9	27.0	24.6	25.5	25.7
Constant	41.0	40.7	42.6	47.0	40.7	43.9
Decreasing	31.0	33.3	30.4	28.4	33.8	30.5
Finger size commonly used	Small, Large	Small, Large	Small, Large	Small, Large	Small, Large	Small, Large
Ripening stage (%)						
Matured green	37.9	33.2	56.3	0.2	42.7	92.7
More yellow than green	46.6	40.6	42.2	11.7	40.2	6.6
All yellow	13.9	19.1	1.3	54.6	13.5	0.3
Yellow flecked with brown	1.7	7.0	0.2	33.5	3.7	0.3
Cooking ingredients	Water, salt	Salt	Vegetable oil, salt, sugar	Vegetable oil, salt	Water, vegetable oil, salt	Water, vegetable oil, salt, soy sauce, spices, tomato, groundnut, meats
Common accompaniments	Various sauces, beans, cassava leaf, meats	Vegetables, fruits (avocado, plums, pears), fish, meats	Beans, eggs, fish, meats	Eggs, fish, meats, vegetable sauces, ugali	Various sauces	-
Cooking time (min) ^1^	33.6 (12.0)	13.0 (8.1)	10.7 (6.7)	10.3 (7.7)	43.6 (20.1)	65.6 (25.7)

^1^ Value is the mean (standard deviation). ^2^ Chips are mostly consumed as a snack, but some also consume them with other accompaniments.

**Table 3 foods-10-01955-t003:** Consumption characteristics and cooking methods of plantain-based products in Nigeria.

Products	Boiled Plantain (*n* = 317)	Roasted Plantain (*n* = 237)	Fried as Chips (*n* = 274)	Fried as *dodo* (*n* = 406)	Flour ^2^ (*n* = 164)	Pounded (*n* = 89)
Consumption frequency (%)						
1 time/day	8.2	7.2	22.6	25.6	12.2	3.4
>1 time/day	4.1	6.8	9.5	8.6	6.1	5.6
Weekly	45.1	37.1	33.6	54.9	34.8	39.3
Monthly	18.3	21.5	13.1	6.4	20.7	22.5
Occasionally	24.3	27.4	21.2	4.4	26.2	29.2
Consumption trend till now (%)						
Increasing	58.9	54.2	60.7	73.9	61.0	51.1
Constant	24.8	24.2	23.3	17.7	23.8	27.3
Decreasing	16.3	21.6	16.0	8.4	15.2	21.6
Finger size commonly used	Large	Small, Large	Small, Large	Large	Small, Large	Large
Ripening stage (%)						
Matured green	64.2	25.0	70.1	1.2	95.7	88.9
More yellow than green	16.5	38.6	29.9	20.9	3.7	7.8
All yellow	18.5	35.2	0.0	75.4	0.6	3.3
Yellow flecked with brown	0.9	1.1	0.0	2.6	0.0	0.0
Cooking ingredients	Water, Salt	Salt, Sugar	Vegetable oil, Salt, Sugar	Vegetable oil, Salt	Garlic, Onion, Vegetable oil, Fish, Meats	Water, Yam, Salt
Common accompaniments	Palm oil, stews, fish, eggs, beans, rice, gari, bread, vegetables	Beans, groundnut, palm oil, rice,	Drinking water, soft drinks	Beans, rice, ugali, gari, eggs, meat, stew	Melon soup, vegetable soup, yam, sweet potato, boiled ripe plantain	Melon soup, vegetable soup, stew sauce, fish, meats, ponmo (a meat from the skin of cows, goats, sheep, etc.)
Cooking time (min) ^1^	14.8 (8.7)	12.1 (6.7)	10.0 (5.6)	8.9 (5.5)	20.0 (12.1) ^3^	31.0 (15.0)

^1^ Value is the mean (standard deviation). ^2^ Plantain flour is often mixed with cassava or yam flour before cooking. ^3^ This cooking time means time for preparation of the plantain flour dishes.

**Table 4 foods-10-01955-t004:** Factors determining consumer’s choices for consumption of plantain-based products in Cameroon.

Product Factors	Cameroon				
	Sex	Age	Education	Household Size	Annual Income ^1^
Nutrition					
Always	0.42 (0.48)	40.51 (14.20)	10.51 (3.02)	5.51 (1.24)	0.75 (0.22)
Sometimes	0.44 (0.49)	40.30 (12.18)	8.12 (2.04)	6.03 (1.02)	0.67 (0.23)
Never	0.45 (0.50)	39.23 (14.41)	7.25 (2.31)	5.67 (0.98)	0.48 (0.12)
F-test	0.58	1.52	1.91 *	0.45	1.13 *
Food safety					
Always	0.46 (0.47)	44.71 (12.72)	11.12 (2.15)	5.08 (1.02)	0.65 (0.18)
Sometimes	0.42 (0.49)	40.85 (10.50)	10.56 (1.57)	5.25 (2.21)	0.62 (0.32)
Never	0.42 (0.50)	41.72 (12.33)	12.15 (3.01)	5.43 (1.17)	0.58 (0.21)
F-test	1.01	0.47	0.08	0.95	0.82
Taste					
Always	0.48 (0.49)	40.24 (12.13)	9.56 (1.82)	6.12 (2.03)	0.45 (0.18)
Sometimes	0.37 (0.42)	41.15 (12.82)	10.03 (2.03)	7.14 (1.98)	0.53 (0.20)
Never	0.32 (0.45)	40.72 (13.50)	9.82 (1.14)	6.57 (2.36)	0.47 (0.34)
F-test	1.52 **	0.05	0.70	1.06	0.81
Appearance					
Always	0.47 (0.50)	39.87 (12.40)	10.25 (2.03)	7.28 (2.11)	0.52 (0.35)
Sometimes	0.39 (0.47)	38.54 (11.35)	9.88 (1.95)	7.54 (2.31)	0.45 (0.51)
Never	0.42 (0.49)	39.23 (10.24)	10.04 (2.56)	7.08 (2.12)	0.48 (0.36)
F-test	0.98	0.53	0.90	1.00	0.07
Availability					
Always	0.41 (0.50)	40.17 (11.82)	11.11 (4.05)	6.02 (2.04)	0.45 (0.32)
Sometimes	0.43 (0.49)	39.91 (13.15)	10.58 (2.03)	5.78 (2.13)	0.37 (0.30)
Never	0.45 (0.49)	39.43 (12.37)	11.02 (2.12)	6.74 (3.01)	0.46 (0.27)
F-test	0.68	0.87	0.96	0.75	0.69
Price					
Always	0.43 (0.49)	38.80 (10.84)	10.12 (2.54)	7.21 (1.15)	0.52 (0.34)
Sometimes	0.42 (0.48)	37.51 (12.31)	9.86 (3.02)	5.23 (1.09)	0.21 (0.18)
Never	0.45 (0.51)	39.23 (11.57)	9.44 (2.25)	4.78 (2.05)	0.38 (0.41)
F-test	0.97	0.00	0.70	1.35*	1.07
Size and quantity					
Always	0.45 (0.50)	39.62 (13.03)	10.23 (2.57)	6.78 (2.31)	0.46 (0.35)
Sometimes	0.47 (0.49)	40.43 (12.46)	11.01 (2.18)	5.22 (1.09)	0.50 (0.29)
Never	0.44 (0.50)	41.28 (13.25)	11.14 (2.21)	5.01 (1.23)	0.42 (0.41)
F-test	1.10	0.72	0.51	0.64	0.41
Package					
Always	0.41 (0.49)	38.83 (12.05)	9.87 (1.89)	6.57 (2.15)	0.44 (0.32)
Sometimes	0.39 (0.50)	39.22 (9.18)	10.23 (2.11)	7.02 (1.89)	0.37 (0.13)
Never	0.38 (0.48)	40.17 (10.56)	10.57 (2.78)	6.98 (3.01)	0.46 (0.35)
F-test	0.78	0.64	0.86	0.53	0.51
Location of produce					
Always	0.38 (0.49)	42.24 (11.05)	11.21 (1.85)	7.18 (2.57)	0.52 (0.23)
Sometimes	0.37 (0.48)	41.18 (12.10)	10.57 (2.22)	7.01 (1.45)	0.48 (0.31)
Never	0.35 (0.49)	43.02 (11.23)	11.48 (2.09)	6.74 (2.08)	0.51 (0.43)
F-test	0.98	0.48	1.01	1.06	0.15
Hunger					
Always	0.47 (0.49)	38.72 (9.82)	12.12 (3.02)	6.47 (1.75)	0.47 (0.34)
Sometimes	0.38 (0.49)	39.13 (12.03)	12.25 (1.92)	6.38 (2.21)	0.41 (0.28)
Never	0.43 (0.50)	39.46 (10.54)	11.89 (2.04)	6.69 (2.05)	0.50 (0.43)
F-test	0.56	0.84	0.84	0.70	1.02
Knowledge					
Always	0.45 (0.49)	45.75 (12.19)	12.08 (2.15)	6.89 (3.00)	0.41 (0.35)
Sometimes	0.47 (0.50)	40.68 (11.14)	11.57 (2.32)	7.18 (2.15)	0.58 (0.44)
Never	0.43 (0.48)	41.36 (9.96)	11.43 (3.11)	6.47 (2.11)	0.52 (0.32)
F-test	1.12	0.92	0.61	1.37	0.73

Standard deviations are in parentheses. *, ** indicate significant differences at levels of 5% and 1%, respectively. ^1^ 1 if <USD 1800, otherwise 0.

**Table 5 foods-10-01955-t005:** Factors determining consumers’ choices for consumption of plantain-based products in Nigeria.

Product Factors	Nigeria				
	Sex	Age	Education	Household Size	Annual Income ^1^
Nutrition					
Always	0.48 (0.48)	40.23 (13.18)	13.57 (4.12)	5.09 (1.18)	0.55 (0.49)
Sometimes	0.42 (0.49)	44.04 (12.56)	10.08 (3.51)	4.57 (0.95)	0.31 (0.45)
Never	0.44 (0.50)	43.10 (11.64)	8.14 (3.23)	4.98 (2.01)	0.24 (0.37)
F-test	0.62	1.72	1.98 **	0.52	1.08 *
Food safety					
Always	0.45 (0.49)	43.54 (11.98)	12.08 (3.07)	4.78 (2.14)	0.48 (0.23)
Sometimes	0.47 (0.50)	44.15 (12.03)	12.56 (2.34)	4.52 (1.55)	0.42 (0.41)
Never	0.41 (0.48)	42.08 (13.11)	10.78 (2.75)	4.39 (1.72)	0.40 (0.43)
F-test	1.28	0.56	0.15	1.05	0.67
Taste					
Always	0.45 (0.49)	45.75 (13.02)	11.16 (2.56)	5.71 (1.87)	0.56 (0.48)
Sometimes	0.35 (0.50)	44.38 (14.15)	12.23 (3.12)	5.14 (2.02)	0.48 (0.50)
Never	0.30 (0.48)	45.69 (13.78)	13.02 (2.45)	5.49 (1.28)	0.52 (0.44)
F-test	1.44 *	0.10	0.87	0.98	0.92
Appearance					
Always	0.42 (0.50)	41.25 (14.49)	12.14 (2.76)	6.01 (2.08)	0.47 (0.40)
Sometimes	0.43 (0.50)	41.36 (15.14)	11.47 (2.85)	5.78 (1.98)	0.42 (0.48)
Never	0.44 (0.49)	40.68 (12.86)	11.87 (3.03)	6.13 (2.32)	0.45 (0.39)
F-test	1.08	0.44	0.97	0.87	0.11
Availability					
Always	0.43 (0.48)	41.23 (12.05)	13.42 (5.02)	4.87 (1.95)	0.38 (0.35)
Sometimes	0.44 (0.50)	42.18 (11.57)	12.28 (3.12)	5.13 (2.04)	0.40 (0.41)
Never	0.40 (0.49)	41.84 (13.08)	12.51 (2.87)	4.58 (2.29)	0.39 (0.48)
F-test	0.51	0.95	1.03	0.87	0.78
Price					
Always	0.40 (0.49)	39.67 (11.45)	12.03 (2.05)	6.28 (1.19)	0.54 (0.52)
Sometimes	0.40 (0.50)	40.08 (12.08)	12.28 (3.24)	5.02 (2.01)	0.42 (0.29)
Never	0.44 (0.45)	41.13 (10.92)	13.04 (3.09)	4.19 (1.28)	0.50 (0.32)
F-test	1.03	0.05	0.67	1.47*	1.11
Size and quantity					
Always	0.41 (0.50)	44.08 (12.15)	12.08 (3.26)	4.98 (1.75)	0.52 (0.50)
Sometimes	0.43 (0.49)	42.15 (13.11)	13.19 (2.54)	5.26 (2.02)	0.55 (0.41)
Never	0.45 (0.50)	43.16 (10.75)	12.59 (2.78)	5.14 (1.92)	0.48 (0.28)
F-test	0.89	0.58	0.63	0.60	0.37
Package					
Always	0.35 (0.50)	41.68 (12.65)	11.25 (2.43)	5.28 (2.04)	0.53 (0.47)
Sometimes	0.32 (0.50)	40.15 (10.08)	12.09 (3.16)	5.02 (2.22)	0.48 (0.50)
Never	0.37 (0.48)	39.48 (12.13)	12.72 (2.49)	6.13 (1.75)	0.50 (0.42)
F-test	0.84	0.86	0.65	0.57	0.64
Location of produce					
Always	0.40 (0.49)	44.25 (9.87)	12.09 (3.51)	5.45 (1.67)	0.35 (0.32)
Sometimes	0.37 (0.50)	46.12 (11.68)	13.61 (2.52)	5.87 (2.11)	0.43 (0.41)
Never	0.39 (0.50)	43.08 (10.56)	12.58 (2.71)	5.13 (1.89)	0.39 (0.47)
F-test	1.15	0.55	1.20	1.14	0.09
Hunger					
Always	0.42 (0.50)	45.05 (12.11)	14.03 (1.45)	5.08 (2.03)	0.42 (0.42)
Sometimes	0.40 (0.49)	44.21 (11.45)	14.11 (2.05)	4.89 (1.14)	0.50 (0.35)
Never	0.39 (0.48)	42.09 (13.01)	13.87 (2.44)	5.12 (1.97)	0.44 (0.47)
F-test	0.45	0.76	0.92	0.56	0.98
Knowledge					
Always	0.37 (0.50)	46.12 (10.56)	13.42 (3.45)	5.87 (2.05)	0.51 (0.40)
Sometimes	0.32 (0.50)	44.08 (12.08)	12.57 (2.87)	5.13 (2.23)	0.50 (0.44)
Never	0.35 (0.49)	45.04 (10.76)	14.23 (2.98)	5.46 (1.74)	0.55 (0.39)
F-test	1.28	0.83	0.75	1.51	0.82

Standard deviations are in parentheses. *, ** indicate significant differences at levels of 5% and 1%, respectively. ^1^ 1 if <USD 1800, otherwise 0.

**Table 6 foods-10-01955-t006:** Respondents’ satisfaction and attitude responsible for plantain products.

No.	Statements	Response (%)		Chi-Square Value	*p* Value
		Cameroon	Nigeria		
1	I have purchased plantain products from the market.	78.5	94.5	4.59	0.01
2	I am very satisfied with those plantain products from the market.	70.1	87.6	3.25	0.04
3	I would like to pay slightly more for new plantain products if their quality can improve my health.	99.3	98.1	3.02	0.38
4	I would buy new plantain products if they are more delicious.	76.2	66.2	5.14	0.12
5	I would buy new plantain products if they are more nutritious.	61.9	78.5	6.13	0.05
6	I would buy new plantain products if there is a greater quantity.	30.2	25.0	6.78	0.21
7	I would buy new plantain products if they have a lower price.	47.6	44.6	5.45	0.28
8	I would buy new plantain products if they are available in the market.	54.3	52.2	7.52	0.32

## Data Availability

This manuscript has no associated data.

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
