# Peer review of "Consumer Preferences and Socioeconomic Factors Decided on Plantain and Plantain-Based Products in the Central Region of Cameroon and Oyo State, Nigeria"

_foods, 2021, doi:10.3390/foods10081955_

Round 1
Reviewer 1 Report
The authors need to follow the following instructions to improve this manuscript.
- The authors should concisely rewrite the abstract based on the best findings with significant value.
- Line 44 (these): The authors should mention the bioactive compounds.
- Line 52-53: This sentence data should update up to recent years with reference.
- Line 65-66: This sentence data should update up to recent years with reference.
- Table 1: Why do we not use standard deviation in all data?
- Table 4/5 (Food safety): How the authors measured food safety?
- Line 346-347: How do the authors understand that taste and nutrients are key concerns?
- Results and discussions should improve by compare and contrast.
- Line 394: The authors should check recent data
- The conclusion should precisely write based on the best findings.
- The authors should check the English.
Author Response
Manuscript: foods-1334020
Title: Consumer preferences and socioeconomic factors decided on plantain and plantain-based products in the Central Region of Cameroon and Southwestern region of Nigeria
We would like to thank the editor and reviewers for their helpful, constructive, and pertinent suggestions and comments, which we believe have helped to improve quality and scientific substance of the manuscript. The responses have been shown in the yellow highlights of the manuscript.
Response to Reviewer 1:
R 1.1: The author should concisely rewrite the abstract based on the best findings with significant value.
The abstract has been rewritten.
R 1.2: Line 44 (these): The authors should mention the bioactive compounds.
To cover those nutrients (dietary fiber, minerals, vitamins, and phenolic compounds) in the first sentence, the words ‘these bioactive compounds’ have been changed to ‘those chemical constituents’ as presented in Lines 51-52.
R 1.3: Line 52-53: This sentence data should update to recent years with reference.
The data has been updated.
R 1.4: Line 65-66: This sentence data should update to recent years with reference.
We were trying to check the data from FAOSTAT, unfortunately there is no data available for plantain processing after a year 2013.
R 1.5: Table 1: Why do not we use standard deviation in all data?
In Table 1, it could be seen that some data had been collected in number like age, household size, etc, therefore we could present them by average value with standard deviation, while some data had been collected by code. For example, rural = 1, urban = 2, and peri-urban = 3, by this the data was calculated in percentage (with no standard deviation).
R 1.6: Table 4/5 (Food safety): How the authors measured food safety?
In this part, the respondents were asked to rank the importance of these factors (nutrition, food safety, taste, appearance, availability, price, size and quantity, packaging, advertisement, location of produce, and knowledge) on their choice to consume plantain and their products (Always = 1; Sometimes = 2; Never = 3). For ‘food safety’, the definition as ‘food (plantain-based products) that will not harm the consumer when it is prepared and/or consumed in accordance with the food’s purpose’. This had been added in the section 2.1.
R 1.7: Line 346-347: How do the authors understand that taste and nutrients are key concerns?
In section 3.3, Lines 249-255: It could be seen that taste was the most important factor affecting the choice for consuming fresh plantain (66%) and their products (81%) in Cameroon. On the other hand, nutrition was the most positively important factor influencing Nigerian consumer’s choice in consuming fresh plantain (69%) and their products (68%). However, this part has been removed from the conclusion.
R 1.8: Results and discussions should improve by compare and contrast.
Results and discussions have been improved.
R 1.9: Line 394: The authors should check recent data.
The data has been checked and provided for plantain production, however as aforementioned there is no recent data available for plantain processing in Cameroon.
R 1.10: The conclusion should precisely write based on the best findings.
The conclusion has been rewritten.
R 1.11: The authors should check the English.
The English has been checking throughout the manuscript.
Reviewer 2 Report
The paper “Consumer preferences and socioeconomic factors decided on plantain and plantain-based products in Cameroon and Nigeria” contributes to the growth of literature for nutritionists as well as food producers offering plantain and plantain-based products.
the following items should be revised:
The analysis is based on a household survey sample of 454 Cameroonian consumers in four divisions of Central Region and 418 Nigerian consumers in seven government areas of Oyo State.
The population of Cameroon is 27,272,613 and of Nigeria is 211,758,940
This tested group does not represent the entire Cameroon and Nigeria community. That is why I suggest including in the Title: selected regions
The goal of this paper is not clearly emphasized. Maybe better a clear definition of the aim. This part of the text similar to the section of Methods.
Line 29 “Taste- and nutrition were identified as the most important factors that influenced consumers’ choice in consuming plantain and its products”
Has been tested the preferences for the types of preferred flavors (descriptors)?
Introduction
The authors did not write on the nutritional value of plantain and plantain-based products. Why consumption of this product is important.
The part of Table 1 and point 3.1 should be part of the methodology for example "Location of the household"
Conclusions
Line 346 “As taste and nutrient content are key concerns among consumers, this information is useful to producers and breeders.” How do the results respond to the sentence? The authors did not show the preferences for the types of preferred flavors.
Line 347 “innovative approaches” what innovative?
The Conclusion is similar to the results summary.
Author Response
Manuscript: foods-1334020
Title: Consumer preferences and socioeconomic factors decided on plantain and plantain-based products in the Central Region of Cameroon and Southwestern region of Nigeria
We would like to thank the editor and reviewers for their helpful, constructive, and pertinent suggestions and comments, which we believe have helped to improve quality and scientific substance of the manuscript. The responses have been shown in the yellow highlights of the manuscript.
Response to Reviewer 2:
R 2.1: The analysis is based on a household survey sample of 454 Cameroonian consumers in four divisions of Central Region and 418 Nigerian consumers in seven government areas of Oyo State. The population of Cameroon is 27,272,613 and of Nigeria is 211,758,940. This tested group does not represent the entire Cameroon and Nigeria community. That is why I suggest including in the Title: selected regions
The title has been changed.
R 2.2: The goal of this paper is not clearly emphasized. Maybe better a clear definition of the aim.
The goal in the introduction part has been improved.
R 2.3: Line 29 “Taste- and nutrition were identified as the most important factors that influenced consumers’ choice in consuming plantain and its products”. Has been tested the preferences for the types of preferred flavors (descriptors)?
Flavors were not used as a factor in consuming plantains and their based-products in this study. However, the sentence in Line 29 has been changed as presented in Lines 31-33.
R 2.4: The authors did not write on the nutritional value of plantain and plantain-based products. Why consumption of this product is important.
The nutrition value of plantain has been given in Lines 46-51.
R 2.5: The part of Table 1 and point 3.1 should be part of the methodology for example "Location of the household".
These parts have been moved to the methodology as suggested.
R 2.6: Line 346 “As taste and nutrient content are key concerns among consumers, this information is useful to producers and breeders.” How do the results respond to the sentence?
These sentences have been revised as presented in Lines 408-410.
R 2.7: Line 347 “innovative approaches” what innovative?
Some examples of innovative approaches have been given to the conclusion part as presented in Lines 427-429.
R 2.8: The Conclusion is similar to the results summary.
The conclusion has been improved.
Round 2
Reviewer 1 Report
No further comments